# Topical Application of Chlorhexidine Gel with Brush-On Technique in the Tailored Treatment of Plaque Induced Gingivitis

**Gianna Maria Nardi [1], Roberta Grassi [2], Guglielmo Campus [3] , Maria Letizia Pareti [1], Dario Di Stasio [4] , Massimo Petruzzi [5] and Fedora della Vella [5,\*]**

1   Department of Oral and Maxillo-Facial Sciences, University of Rome "La Sapienza", 00185 Rome, Italy; giannamaria.nardi@uniroma1.it (G.M.N.); m.lety@live.it (M.L.P.)
2   Department of Surgical, Medical and Sperimental Scinces, University of Sassari, 07100 Sassari, Italy; grassi.roberta93@gmail.com
3   Department of Restorative, Preventive and Pediatric Dentistry, University of Bern, 3010 Bern, Switzerland; guglielmo.campus@zmk.unibe.ch
4   Multidisciplinary Department of Medical-Surgical and Odontostomatological Specialities, University of Campania "Luigi Vanvitelli", 80138 Naples, Italy; dario.distasio@unicampania.it
5   Interdisciplinary Department of Medicine, University of Bari "Aldo Moro", 70124 Bari, Italy; massimo.petruzzi@uniba.it
\*   Correspondence: dellavellaf@gmail.com

**Abstract:** Aim: This study aimed to assess the action of a chlorhexidine-based brush-on gel application in the treatment of plaque-related gingivitis. Methods: The enrollment involved consecutive patients diagnosed with plaque-induced gingivitis. Each participant's full mouth plaque score and gingival index were recorded at the first appointment (t0) and at follow ups after 1 week (t1), 2 weeks (t2) and 3 weeks (t4). All patients were randomly sorted into two groups: A study group, who was given instruction to brush their gums daily with a chlorhexidine gel, and a control group who received a placebo gel. The two groups' data at baseline were compared using a chi-square test, while the t-Student and Mann–Whitney tests were employed to analyze the index's trends, both separately and compared. Results: In total, 30 patients were enrolled, 15 referring to the study group and 15 to the control group. The study group had an 87% decrease in their plaque score and an 84% decrease in their gingival index ($p < 0.05$). The control group displayed a reduction in plaque score and gingival index of 74% and 84%, respectively ($p < 0.05$). The plaque score decreased statistically more significantly in the study group than in the control group. **Conclusion**: Topical employment of a chlorhexidine-containing brush-on gel appears to be a useful home tool in the treatment of plaque-induced gingivitis, associated with professional debridement.

**Keywords:** gingivitis; chlorhexidine; oral hygiene; topical gel

---

## 1. Introduction

Dental plaque-associated gingivitis is the most common periodontal disease, with a prevalence of 30–54% [1,2]. This gingival inflammatory condition is triggered by bacteria accumulation, and it is characterized by clinical features such as swollen, tender, erythematous and bleeding gums, halitosis and pain. The inflammation is found from the free attached gingiva to the mucogingival junction, and according to the number of sites involved it can be classified as incipient, localized or generalized [3]. This is a reversible disease that can lead to a restitutio ad integrum or evolve into periodontitis, depending on the removal of the main etiological factor: the bacterial biofilm.

Periodontitis is a chronic irreversible degenerative disease, involving the periodontal connective tissues and the bone, and inducing teeth attachment loss [4,5]. Even though dental plaque is the necessary key factor for gingivitis development, the severity of the disease is also related to oral and systemic conditions, such as hormones, pharmacological treatment, smoking, hyposalivation and dental restorations [3,6]. Poor oral hygiene is a negative prognostic factor, regardless of other concurrent conditions, thus an adequate plaque control and removal of microbial biofilm is the essential therapeutic measure to adopt in gingivitis management [3,7]. The non-surgical periodontal treatment is based on mechanical debridement, i.e., root plaining and scaling, eventually implemented with antibacterial and antiseptic adjuvants, such as triclosan, locally administrated tetracyclines and chlorhexidine (CHX) [8–10]. Other chemical agents had been proven to be effective in reducing subgingival bacteria presence and activity, including thymol, xylitol and sodium fluoride [11,12]. CHX is one of the most widely used antiseptics, mainly employed for skin and mucosa disinfection, active against Gram-positive and Gram-negative bacteria, capsulated virus and fungi. CHX digluconate-added mouthwashes, oral gels and toothpastes are available as home-based anti-plaque tools. This study evaluated the effect of the daily use of an oral brush-on gel containing CHX digluconate (0.2%), xylitol and sodium fluoride (0.2% NaF) on plaque index and gingival index in patients affected by dental plaque-induced gingivitis.

## 2. Materials and Methods

A non-probability consecutive sampling procedure was used within 12 weeks, from January to March 2018, among patients attending a dental clinic in Rome (Italy) with a diagnosis of plaque-induced gingivitis, according to the American Academy of Periodontology (AAP) guidelines [2]. The sample size was estimated using a finite population correction (FPC) considering the Lazio population with a mean age assimilable to our sample and a gingivitis prevalence of 20%, according to AAP 2017 workshop data [2].

Each patient underwent an anamnestic and clinical examination after signing an informed consent to study participation. The study was conducted in accordance with the Declaration of Helsinki of 1975 and was approved by the local Ethical Committee (prot. nr. 1216/19). The exclusion criteria were pregnancy, presence of hematologic pathologies, current anticoagulant therapy, presence of diagnosed autoimmune gum diseases and refusal of study participation.

All patients' full mouth plaque score (FMPS) and gingival index (GI) [13] were measured and recorded at the first visit (t0). FMPS was performed by applying a fluorescein plaque detector and using VALO LED curing light, according to the Dental Biofilm Detection Topographic Technique (D-Biotech) protocol, in order to achieve a more accurate and ergonomic deplaquing. Patients were randomly divided into a study group and a control group; both received full mouth scaling and were educated as concerns a home oral hygiene method based on the Tailored Brushing Method (TBM) [14]. The study group received indication to brush their teeth once a day without time specification using a full brush (1 cm, approx. 0.5 g) dental gel with chlorhexidine digluconate (0.2%) (Cervitec Gel, Ivoclar Vivadent, Schaan-Liechtestein), and to avoid drinking and eating for at least 30 minutes after the application, while the control group were provided with a chlorhexidine-free gel (placebo). Both gels contained sodium fluoride (0.2%). The two gels were identical in taste, color and texture. Neither the researcher nor the patients were aware of the group allocation.

Every patient was asked to not use any kind of other toothpaste or mouthrinse and was told to report any adverse effect.

FMPS and GI were measured after 1 week (t1), 2 weeks (t2) and 3 weeks (t3) by the same operator (GMN).

*Statistical Analysis*

The two groups' demographical data were compared using chi-square test, while the groups' FMPS and GI values at t0, t1, t2 and t3 were compared using a t-Student test, with statistical significance

set at <0.05. A two-tailed Mann–Whitney test was employed to evaluate the statistical significance of the FMPS and GI values' difference from t0 to t3 within both groups.

## 3. Results

In total, 30 patients were enrolled, 17 females and 13 males, with a mean age of 48 years (SD: ± 14): 15 referred to the study group and 15 to the control group. In total, six patients dropped out: three patients from the study group did not attend the follow-up visits, one patient from the control group missed the follow-ups and two patients from the control group did not adhere to the study protocol, and therefore were excluded from the study. The patients' enrollment flowchart is shown in Figure 1. The study and the control group were statistically comparable for demographical data, such as for baseline FMPS and GI ($p > 0.05$). The study group showed an FMPS improvement of 87% ($p < 0.05$) and a GI decrease of 84% ($p < 0.05$), while the control group showed an FMPS and GI decrease of 75% ($p < 0.05$) and 80% ($p < 0.05$), respectively.

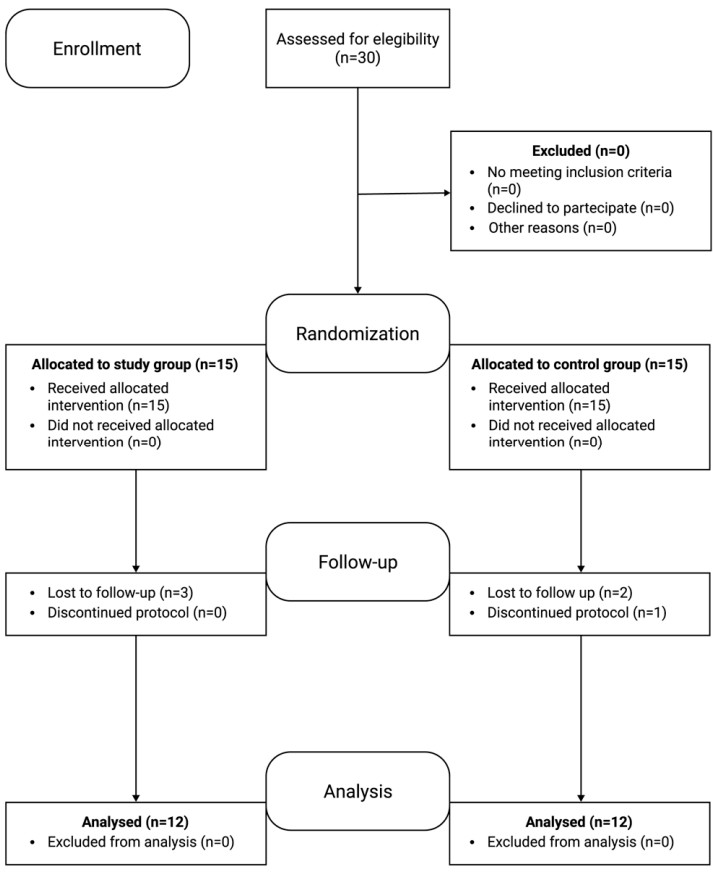

**Figure 1.** Flow-chart of the study enrollment.

Both the study and the control group showed a statistically significant reduction ($p < 0.03$) of FMPS and GI indexes from t0 to t3 (Table 1).

**Table 1.** Full mouth plaque score (FMPS) and Gingival index (GI) reduction from t0 to t3 in study and control groups.

| | Group | n. Patients | Decrease t0–t3 (%) | *p* Value |
|---|---|---|---|---|
| FMPS | Study | 12 | 87% | 0.03 * |
| | Control | 12 | 75% | 0.03 * |
| GI | Study | 12 | 84% | 0.03 * |
| | Control | 12 | 80% | 0.03 * |

* Statistically significant ($p < 0.05$).

No statistically significant differences between the clinical indexes of the two groups emerged at t1, t2 or t3, while the FMPS index improvement ($\Delta$(t0–t3)) appeared to be significantly better in the study group (Table 2)

**Table 2.** Comparison of clinical indexes for test group and control group.

| | Patients | | | Plaque Score (Means) | | | | | Gingival Index (Means) | | | | |
|---|---|---|---|---|---|---|---|---|---|---|---|---|---|
| | n | Mean Age | sex | t0 | t1 | t2 | t3 | Δ(t0–t3) | t0 | t1 | t2 | t3 | Δ(t0–t3) |
| Study group | 12 | 50 | F = 7 M = 5 | 2.11 SD = 0.6 | 1.09 SD = 0.3 | 0.63 SD = 0.2 | 0.31 SD = 0.1 | 1.83 SD = 0.5 | 1.6 SD = 0.5 | 0.9 SD = 0.3 | 0.54 SD = 0.2 | 0.26 SD = 0.1 | 1.34 SD = 0.4 |
| Control group | 12 | 46 | F = 6 M = 6 | 1.16 SD = 0.5 | 0.58 SD = 0.4 | 0.46 SD = 0.4 | 0.48 SD = 0.3 | 0.87 SD = 0.3 | 1.87 SD = 1.1 | 1.46 SD = 1.4 | 0.87 SD = 0.5 | 0.68 SD = 0.5 | 1.17 SD = 0.6 |
| *p* value | | 0.21 | | 0.68 | 0.16 | 0.5 | 0.5 | 0.42 | **0.049** | 0.13 | 0.34 | 0.48 | 0.29 | 0.17 |

In bold the significant values. SD: Standard deviation.

The study and control group outcomes are summarized in Table 2. None of the patients reported adverse events or side effects during the study.

## 4. Discussion

Plaque-related gingivitis is the most common infective oral disease together with caries [15]; its proper treatment and management is crucial in the prevention of chronic periodontitis, which is the main cause of tooth loss, with negative consequences on life quality and global health [16].

This study highlighted the value of a CHX gel as an adjuvant treatment for patients following non-surgical periodontal treatment. Patients who applied the gel daily ended up with larger improvements of both FMPS and GI, compared with the control group.

The FMPS underwent a greater and quicker improvement from the first week of use, while the GI decrease required more time. This is probably due to the character of the indexes: in fact, the GI is a tissue inflammation parameter, valuing the degree of gingival injury, and therefore the gingival epithelium restoration can only happen after the removal of the irritative bacterial trigger. In addition, GI is more easily affected by individual factors, like gingival biotype, age, smoking and teething [3].

The plaque accumulation's removal and control are essential for a successful periodontal disease therapy, and professional oral hygiene instructions are proven to boost the gingival inflammation indexes [17]. The utility of additional chemical anti-plaque agents for gingivitis treatment is acknowledged [18], and the active substance delivery format is relevant to the patient needs and compliance. The use of CHX mouthwashes appears to be more effective than toothpastes, while, when compared to gel and varnish, mouthwashes were less powerful in plaque control [19]. The gel ensures a longer-term release and antibacterial activity of CHX, while fluid formulation only demonstrated a short-term effect [19]. This is probably due to the higher control that the patient has over the gel application, which allows them to perfectly dose the amount of gel needed in every site and guarantees that the CHX contained in the product reaches all the desired areas, while liquid textures are less controllable. Moreover, the gel can stay longer on the gums, since the patients were asked to avoid drinks, food and rinses for at least 30 minutes after application, while mouthwashes are generally kept in the mouth only for a few minutes and often rinsed with water.

The brush-on application of CHX brings the benefit of a lower risk of teeth discoloration, which is the main side effect of this antiseptic compared to mouthwashes [20], other than the easier compliance for the patients, requiring a gesture similar to teeth brushing.

The synergy with the xylitol contained in the gel formulation allows an enhancing of the anti-inflammatory and anti-bacterial effects, thanks to xylitol's inhibition of cytokines release, bacteria attachment and Nitric Oxide (NO) production [12,21].

The efficacy of this kind of domiciliary protocol has already been demonstrated in the elderly, who are at higher risk of periodontal infective pathology development, and in patients affected by peri-implant mucositis [22,23].

## 5. Conclusions

From this study, the combination of professional mechanical debridement, which remains the gold standard, with a home self-applied CHX gel proved to be useful in improving the clinical parameters for patients affected by plaque-induced gingivitis. Further studies are needed in order to better investigate the long-term effects of this chemical adjuvant and its efficacy in preventing periodontal diseases.

**Author Contributions:** Conceptualization, M.P. and G.M.N.; methodology, G.M.N., M.P., M.L.P.; software, G.C., F.d.V., D.D.S.; validation, R.G., G.M.N., M.P., G.C.; formal analysis, G.C., F.d.V., D.D.S.; investigation, G.M.N., M.L.P., R.G.; resources, G.M.N.; data curation, F.d.V., G.C., D.D.S.; writing—original draft preparation, F.d.V.; writing—review and editing, F.d.V., M.P., G.C.; visualization, R.G., G.M.N.; supervision, G.M.N., G.C., M.P.; project administration, G.M.N., M.P.; funding acquisition, G.M.N. All authors have read and agreed to the published version of the manuscript.

**Funding:** This research received no external funding.

**Conflicts of Interest:** The authors declare no conflict of interest.

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
