# Peer review of "Topical Application of Chlorhexidine Gel with Brush-On Technique in the Tailored Treatment of Plaque Induced Gingivitis"

_applsci, doi:10.3390/app10176014_

Round 1

Reviewer 1 Report

Good study, minor grammar corrections are needed:

Line 20: enrolment, change to enrollment

Line 24: whose, change to who

Line 32: employ, change to employment

Line 45: on, change to to

Line 120: end, change to ended

Author Response

Dear Editor,

I thank you on behalf of all the Authors for the opportunity to review our study.

We adjusted and corrected our manuscript according to the referees’ suggestions, and we firmly believe that these changes improved and added value to our study. All the additions and corrections in the manuscript are highlighted in blue.

To follow a response point by point to the reviewers’ concerns.

Reviewer 1

Good study, minor grammar corrections are needed:

Reviewer: Line 20: enrolment, change to enrollment

Our response: done

Reviewer Line 24: whose, change to who

Our response: done

Reviewer: Line 32: employ, change to employment

Our response: done

Reviewer: Line 45: on, change to to

Our response: done

Reviewer: Line 120: end, change to ended

Our response: done

Reviewer 2 Report

Methods: The patients were instructed to brush their teeth once a day. Please indicate if that's in the morning or at night, or no time specification.

Results: Six patients dropped out: 3 from the study group and 3 from the control group, 5 patients did not attend the follow-up visit, while the remaining did not adhere to the protocol prescriptions. After reading Figure 1, I still cannot be sure what this sentence means. Please rephrase. In Figure 1, "analysis" was spelt wrong.

The study group and the control group did show a statistically significant reduction of FMPS and GI from t0 to t3 (table 1). Please rewrite this sentence. Do you mean there is no significant group difference?

Table 2: It is hard to understand that there was no significant group differences at t0, t1, t2, t3, but the delta(t0-t3) suddenly showed significant group difference. Please report more data (SD or 95% CI) and the exact p value.

Discussion: It is said that CHX gel is more powerful than CHX mouthrinse because the gel ensures a longer-term release and antibacterial activity. Please further clarify: how the patients were instructed to rinse the CHX gel (toothpaste) after brushing to ensure a longer-term release? The context of applying gel/varnish is different from brushing - as we instruct the patient not to eat or drink for ~30 mins after the gel/varnish application, whereas patient will rinse everything away after brushing.

Author Response

Dear Editor,

I thank you on behalf of all the Authors for the opportunity to review our study.

We adjusted and corrected our manuscript according to the referees’ suggestions, and we firmly believe that these changes improved and added value to our study. All the additions and corrections in the manuscript are highlighted in blue.

To follow a response point by point to the reviewers’ concerns.

Reviewer: Methods: The patients were instructed to brush their teeth once a day. Please indicate if that's in the morning or at night, or no time specification.

Our response: dear reviewer, patients did not receive specific instructions about the application time, they only were recommended to avoid drinking and eating for the next 30 minutes to the application, in order to let the gel act. We specified it in the method section.

Reviewer: Results: Six patients dropped out: 3 from the study group and 3 from the control group, 5 patients did not attend the follow-up visit, while the remaining did not adhere to the protocol prescriptions. After reading Figure 1, I still cannot be sure what this sentence means. Please rephrase.

Our response:  We reformulated the sentence as follows: “Six patients dropped out: 3 patients from the study group did not attend the follow-up visits, 1 patient from the control group missed the follow-ups and 2 patients from the control group did not adhere to the study protocol and therefore were excluded from the study”

Reviewer: In Figure 1, "analysis" was spelt wrong.

Our response: Figure 1 was corrected

Reviewer: The study group and the control group did show a statistically significant reduction of FMPS and GI from t0 to t3 (table 1). Please rewrite this sentence. Do you mean there is no significant group difference?

Our response: Dear reviewer, we rewrote the sentence. The two groups FMPS and GI indexes decreased both from t0 to t3. At each time the two groups indexes values were comparable (no statistically significant differences) but the magnitude of the FMPS decrease was significantly higher (p<0.05) for the study group.

Reviewer: Table 2: It is hard to understand that there was no significant group differences at t0, t1, t2, t3, but the delta(t0-t3) suddenly showed significant group difference. Please report more data (SD or 95% CI) and the exact p value.

Our response: The p values and the standard deviation values were included in the table as requested.

Reviewer: Discussion: It is said that CHX gel is more powerful than CHX mouthrinse because the gel ensures a longer-term release and antibacterial activity. Please further clarify: how the patients were instructed to rinse the CHX gel (toothpaste) after brushing to ensure a longer-term release? The context of applying gel/varnish is different from brushing - as we instruct the patient not to eat or drink for ~30 mins after the gel/varnish application, whereas patient will rinse everything away after brushing.

Our response: The patients were told to not eat or drink for at least 30 minutes from the application, in order to allow a prolonged action. Mouthwashes generally stay in the oral cavity for few minutes or seconds, and the fluid texture do not ensure that CHX reaches the interested spots in adequate quantity and for enough time. The gel application is way more controlled and allows to dose the amount of medication needed in every site. This aspect was furthered in the discussion section at line 139.

Reviewer 3 Report

This article is is of huge interest to readers. The quality of the presentation is high. Although the results and conclusions could have been a little bit more clearly represented, I have approved the article without further edits.

Author Response

Dear Editor,

I thank you on behalf of all the Authors for the opportunity to review our study.

We adjusted and corrected our manuscript according to the referees’ suggestions, and we firmly believe that these changes improved and added value to our study. All the additions and corrections in the manuscript are highlighted in blue.

To follow a response point by point to the reviewers’ concerns.

Reviewer: This article is is of huge interest to readers. The quality of the presentation is high. Although the results and conclusions could have been a little bit more clearly represented, I have approved the article without further edits.

Our response: The Authors thank the reviewer for his comment. We hope that the changes made to the results section as requested from reviewer 2 helped to clarify the section.

Round 2

Reviewer 2 Report

In Table 2, delta(t0-t3) has a p value of 0.05. In the methods, it was defined that significant results have p < 0.05. So for Table 2, please give the p value in 3 or more decimal places to show that it is < 0.05. If it equals to 0.05, then it is not significant by definition.

Author Response

Dear reviewer, 

We reported the non-rounded p value in table 2 as requested (0.049).